# Highly efficient anion exchange membrane water electrolyzers via chromium-doped amorphous electrocatalysts

Sicheng Li [1,5], Tong Liu[1,5], Wei Zhang [1 ✉], Mingzhen Wang[2], Huijuan Zhang[1], Chunlan Qin[1], Lingling Zhang[1], Yudan Chen[1], Shuaiwei Jiang[1], Dong Liu [1], Xiaokang Liu[1], Huijuan Wang[3], Qiquan Luo [4], Tao Ding [1 ✉] & Tao Yao [1 ✉]

In-depth comprehension and modulation of the electronic structure of the active metal sites is crucial to enhance their intrinsic activity of electrocatalytic oxygen evolution reaction (OER) toward anion exchange membrane water electrolyzers (AEMWEs). Here, we elaborate a series of amorphous metal oxide catalysts ($FeCrO_x$, $CoCrO_x$ and $NiCrO_x$) with high performance AEMWEs by high-valent chromium dopant. We discover that the positive effect of the transition from low to high valence of the Co site on the adsorption energy of the intermediate and the lower oxidation barrier is the key factor for its increased activity by synchrotron radiation in-situ techniques. Particularly, the $CoCrO_x$ anode catalyst achieves the high current density of 1.5 A cm$^{-2}$ at 2.1 V and maintains for over 120 h with attenuation less than 4.9 mV h$^{-1}$ in AEMWE testing. Such exceptional performance demonstrates a promising prospect for industrial application and providing general guidelines for the design of high-efficiency AEMWEs systems.

Anion exchange membrane (AEM) electrolysis is considered a pivotal technology for the sustainable energy economy[1,2]. It has advantages over proton exchange membrane (PEM) technology with noble metal oxide ($IrO_2$) anodes. These advantages include the ability to use less expensive non-platinum group metal (PGM) catalysts on the anode side, cheaper ion exchange membranes, free of environmentally harmful fluorine-based polymers, and no need for acid-resistant stacking materials, resulting in lower overall device costs[3–6]. Nevertheless, AEMWE technology is a relatively new technology and faces many issues that have to be solved before fulfilling its full potential, with catalysts being a crucial factor[7]. There are two major challenges in developing higher performance non-noble metal anode OER electrocatalysts. One is that current synthesis methods for these catalysts are not optimized for industrial applications. Therefore, in practical scenarios, it is essential to consider simple and easily accessible synthesis

methods that can produce large quantities of catalysts. The other is that although Ni/Fe-based materials have been extensively studied as OER catalysts, their stability is poor and they are rarely used in AEM electrolysis research[8]. The stability and activity are both critical factors especially in the electrolysis applications, which should receive more attention.

On the anode side of AEMWEs, the OER is a complex process involving the transfer of multiple electrons. As the electrons are transferred, the valence state of the active metal changes, which is believed to be the source of the OER intrinsic activity[9]. Hence, there is ongoing debate regarding the relationship between the valence state and activity of the active site in transition-metal-based catalysts (Fig. 1a). Numerous studies have revealed that transition metal compounds with high oxidation state metal sites exhibit great OER activity, while those with low oxidation metal sites have comparatively low

[1]School of Nuclear Science and Technology, Key Laboratory of Precision and Intelligent Chemistry, Hefei National Research Center for Physical Sciences at the Microscale, National Synchrotron Radiation Laboratory, University of Science and Technology of China, Hefei, P.R. China. [2]Zhongke Enthalpy (Anhui) New Energy Technology Co. Ltd, Hefei, P.R. China. [3]Experimental Center of Engineering and Materials Science, University of Science and Technology of China, Hefei, P.R. China. [4]Institutes of Physical Science and Information Technology, Anhui University, Hefei, P.R. China. [5]These authors contributed equally: Sicheng Li, Tong Liu. ✉e-mail: zhangw94@ustc.edu.cn; dingtao@ustc.edu.cn; yaot@ustc.edu.cn

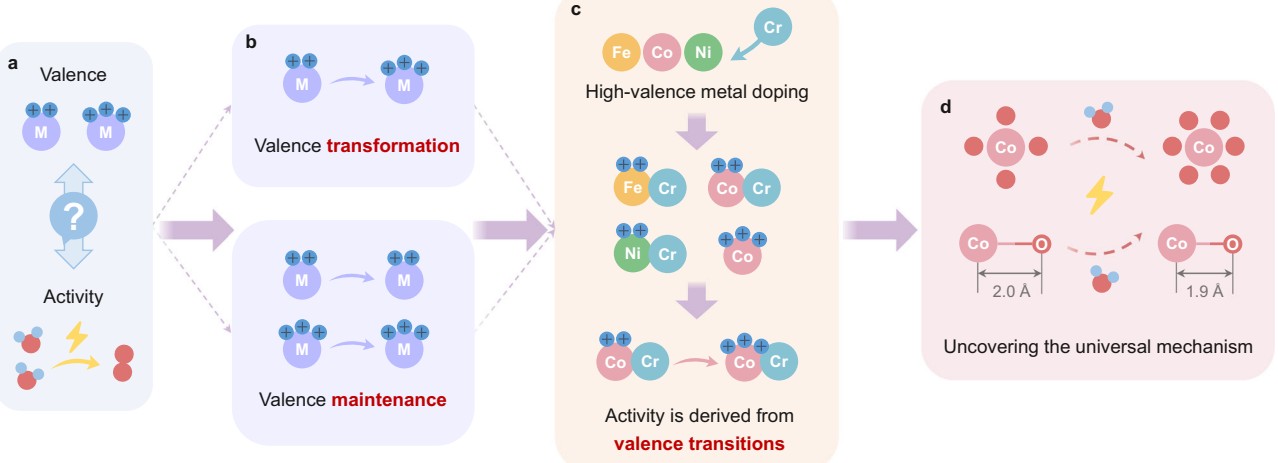

**Fig. 1 | Schematic diagram of the origin of the research and design ideas. a** The inherent debate between active site valence and catalytic activity. **b** Different types of valence changes during the OER. **c** A series of meticulously conceived and designed experiments were conducted using Cr as the high-valence dopant. **d** Uncovering valence changes and universal mechanisms impacting catalytic performance.

activity[10,11]. However, recent research has challenged this notion, suggesting that low oxidation metal sites can also play a significant role in determine OER activity[12-14]. Based on this theory, scientists have introduced oxygen-rich defects[15] and doped high valence metals (such as W[16], Mo[17], Nb[13] and Cr[18]) to lower the valence state of the active metal. Despite these efforts, the relationship between catalytic activity and valence changes during the reaction remains unclear, owing to the lack of the dynamic evolution analysis of active metal sites during the OER condition (Fig. 1b). In-depth understanding and regulation of the electronic structure of the active metal sites in catalysts, as well as their application in practical AEMWEs for mechanistic comprehension, remains a significant challenge. Synchrotron radiation in-situ techniques provide a practical way to address this issue by monitoring the dynamics of the valence state and elucidating the key reasons for its impact on catalytic performance.

In this work, we chose low-cost, abundant Cr with high valence charge as the metal dopant and utilized a one-step liquid phase reduction method to synthesize a series of efficient amorphous metal oxide catalysts for AEMWEs (Fig. 1c). Among them, the $CoCrO_x$ catalyst delivered outstanding activity and stability. We used in-situ valence to core X-ray emission spectroscopy (vtc-XES) for the first time in the electrocatalytic system to precisely identify the ligands of Co, and the σ-interaction strength enhancement of Co-O during the OER process. The combination of in-situ X-ray absorption spectroscopy (XAS), synchrotron radiation infrared (SR-IR) and other experiments along with theoretical calculations allowed us to discern the atomic and electronic structure evolution at the Co site optimizes the adsorption energy of the oxygen intermediate (Fig. 1d), resulting in the low oxidation energy barrier that are crucial for high catalytic activity. Impressively, the AEMWE was assembled by using $CoCrO_x$ as anode catalyst achieved the high current density of $1.5\,A\,cm^{-2}$ at $2.1\,V$ and exhibited outstanding long-term durability for operating over 120 h at $0.5\,A\,cm^{-2}$, further demonstrating its great potential in the application of water electrolysis industry.

## Results
### Synthesis and structural characterization
A diagram of the catalysts synthetic method is shown in Fig. 2a. The catalysts were prepared by a simple one-step liquid-phase reduction method by mixing $NaBH_4$ with the solution of different metal chlorides under stirring, following by washing and drying at room temperature. The presence of $NaBH_4$ facilitated the simultaneous reduction of the metal precursors during the nucleation and growth phases.

This reduction process also allowed for the instantaneous formation of three-dimensional networks through the fusion of metal nuclei[19], enabling the facile preparation of high performance catalysts. The morphology of the prepared catalysts was observed using the scanning electron microscopy (SEM), revealing the porous characteristics of $FeCrO_x$, $CoCrO_x$ and $NiCrO_x$ (Fig. 2b and Supplementary Fig. 1). Moreover, the structural features were investigated by transmission electron microscopy (TEM), which indicated that the prepared catalysts as the disordered atomic structure (Fig. 2c), and spherical nanoparticles with an average size of about 20 nm were observed (Fig. 2d). The high-resolution TEM (HRTEM) and aberration-corrected high-angle annular dark-field scanning transmission electron microscopy (AC HAADF-STEM) further showed that there were no obvious lattice striations (Fig. 2e and Supplementary Fig. 2), indicating its amorphous nature. X-ray diffraction (XRD) pattern (Supplementary Fig. 3) showed very broad and weak diffraction peaks, indicating the long-range disorder feature. The amorphous character is further confirmed by the diffraction rings shown by selected area electron diffraction (SAED) (Fig. 2f)[20,21], with the bright inner rings suggesting the short-range order feature[22]. The morphology and structure of $FeCrO_x$ and $NiCrO_x$ were found to be similar to $CoCrO_x$ (Supplementary Figs. 4–7). In addition, energy-dispersive X-ray spectroscopy (EDS) mapping analysis showed that Co, Cr, and O were uniformly distributed throughout the architecture (Fig. 2g), demonstrating the formation of CoCr oxides rather than phase-separated mixed structures. The same phenomenon was observed for $FeCrO_x$ and $NiCrO_x$ (Supplementary Fig. 8), while the higher O content is attributed to the ease of oxidation of the transition metals. The inductively coupled plasma-atomic emission spectrometer (ICP-AES) was used to further determine the elemental ratios of the prepared catalysts (Supplementary Table 1). The results showed that the atomic ratios of Fe, Co, Ni to Cr were close to 1:1, indicating the precise control of the synthetic method adopted in this study. Furthermore, the extended X-ray absorption fine structure (EXAFS) spectra was used to confirm that the $CoCrO_x$, $FeCrO_x$ and $NiCrO_x$ (Fig. 2h, i and Supplementary Figs. 9–12) catalysts had similar local structures and coordination environments, suggesting that each form structurally similar MCr (M=Fe, Co and Ni) oxides, which lays the basis for further analysis.

### Electrocatalytic OER and AEMWEs performance
The OER activity measurements of the catalysts were conducted in $O_2$-saturated 1 M KOH solution by using a standard three-electrode system. The optimal catalytic performance of the samples was

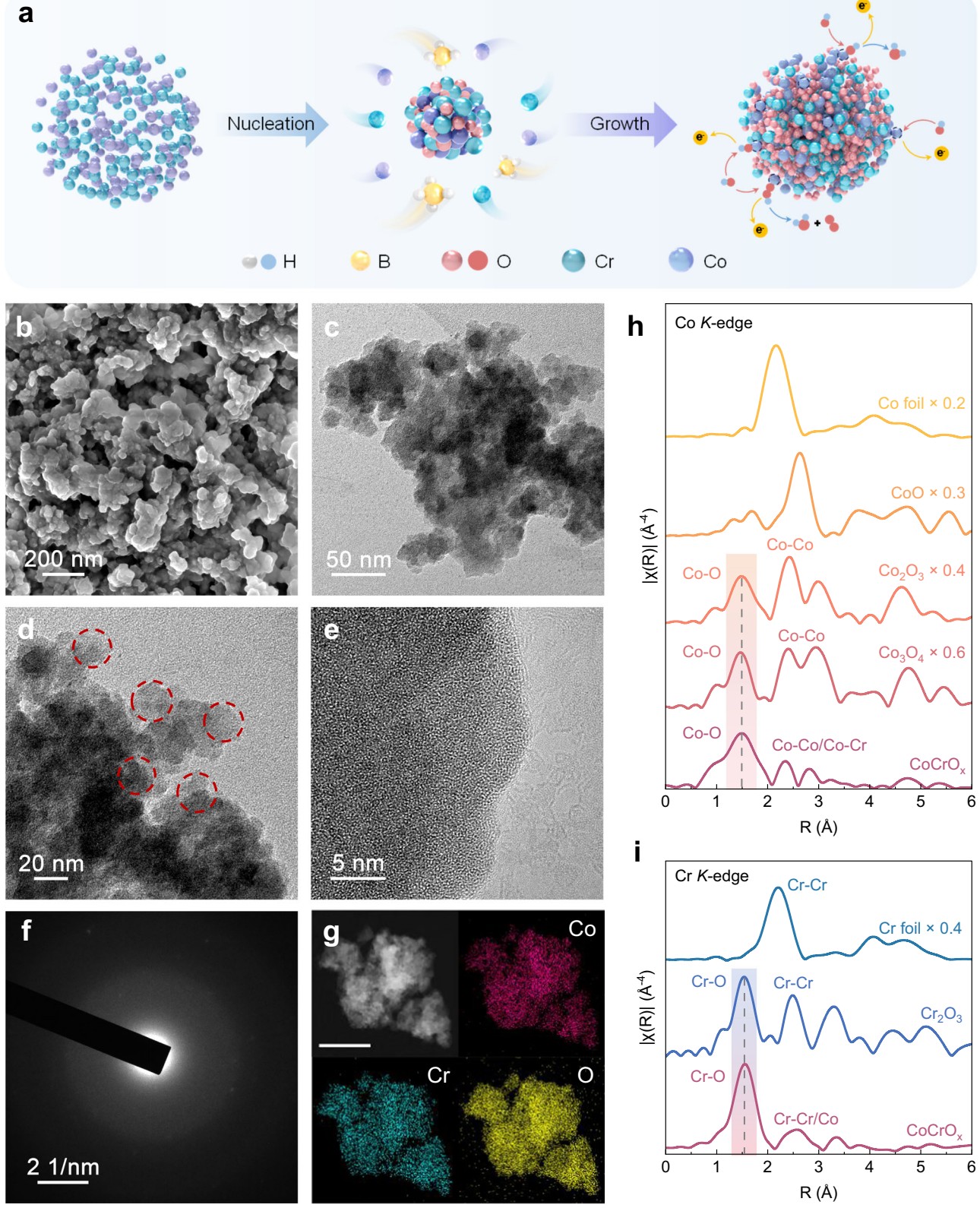

**Fig. 2 | Preparation, structural and morphological characteristics of CoCrO_x.** **a** Illustration of the preparation procedure, (**b**) SEM, (**c**, **d**) TEM, (**e**) HRTEM images. **f** SAED patterns. **g** EDS mappings showing the dispersion of Co (pink), Cr (blue-green) and O (yellow), respectively. Scale bar: 100 nm. EXAFS spectra of (**h**) Co $K$-edge and (**i**) Cr $K$-edge for CoCrO_x and reference samples.

achieved when the atomic ratio of Co/Cr was 1:1, as determined through performance-based screening (Supplementary Fig. 13). For the sake of clarity, the atomic ratio of Fe/Co/Ni to Cr for the FeCrO_x, CoCrO_x and NiCrO_x catalysts is 1:1 unless otherwise specified. To assess the impact of Cr doping, we synthesized a non-Cr doped CoO_x catalyst and also included the benchmark RuO_2 catalyst for comparison. As shown in the iR-corrected LSV curves (Fig. 3a), the CoCrO_x exhibited superior OER performance, necessitating a mere overpotential of

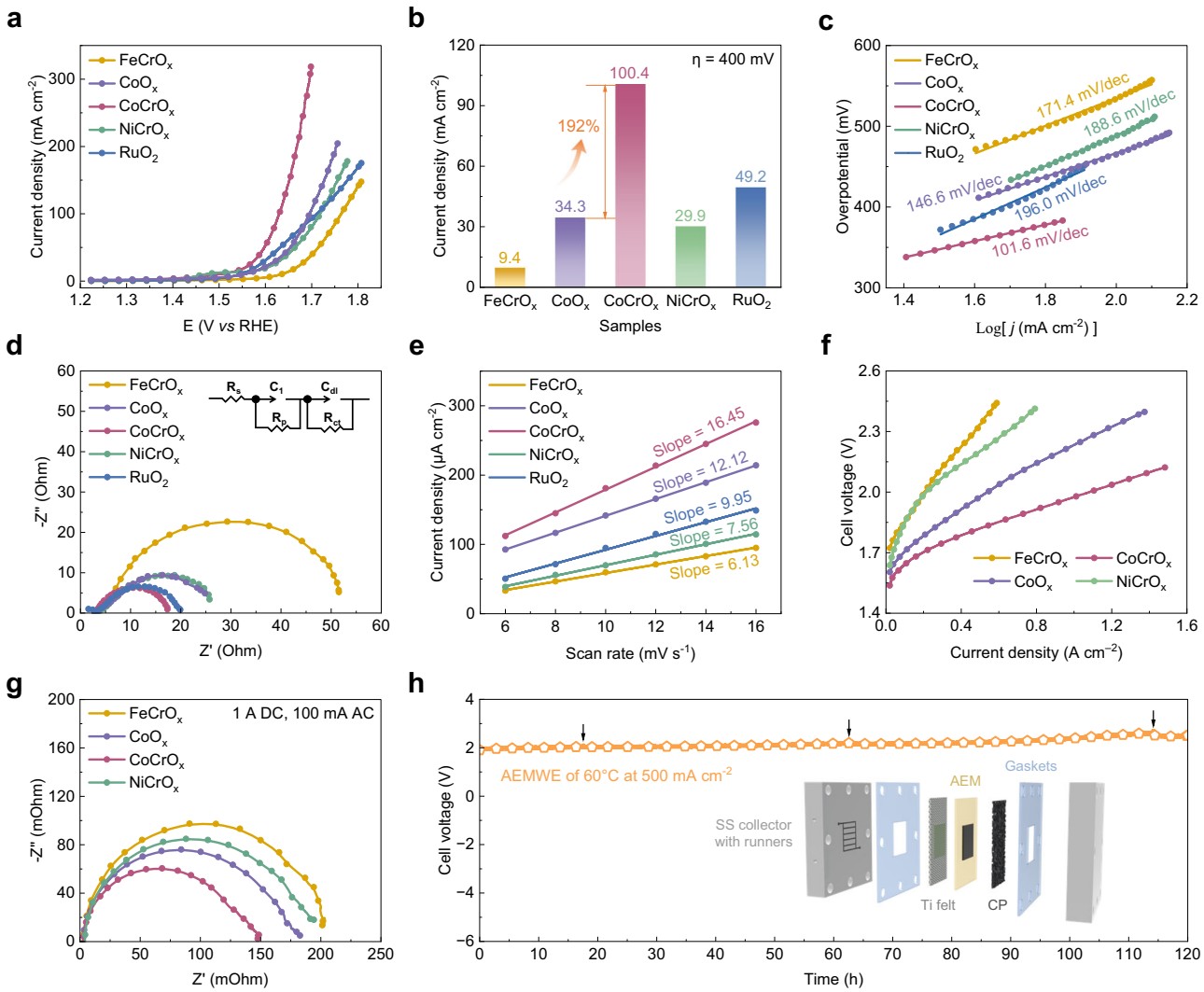

**Fig. 3 | Electrocatalytic activity and stability. a** LSV curves of $FeCrO_x$, $CoCrO_x$, $NiCrO_x$, $CoO_x$ and $RuO_2$ catalysts in 1 M KOH at the scan rate of 5 mV s$^{-1}$. **b** The current density of different catalysts recorded at 400 mV. **c** Tafel plots. **d** Nyquist plots acquired at 1.57 V (vs RHE) in half cell. Inserts in panels d show equivalent circuit of the OER. **e** Differences in current densities plotted against scan rates to determine the ECSA. **f** I–V curves of AEMWEs the prepared catalysts as anodes. **g** Nyquist plots of AEMWEs applying 1 A direct current (DC) and 100 mA root mean square (RMS) of alternating current (AC). **h** Chronopotentiometry curve of $CoCrO_x$ catalyst at constant current density of 0.5 A cm$^{-2}$ in the AEMWEs at 60 °C. Positions indicated by the black arrows represent that the replacement of electrolyte.

268 mV to achieve the current density of 10 mA cm$^{-2}$. The performance outmatched that of other prepared catalysts and the comparative $CoO_x$ catalyst (323 mV), surpassing even the commercial $RuO_2$ (314 mV). It suggested that the beneficial impact of Cr doping on enhancing the OER activity. As the overpotential was fixed at 400 mV, the current density of $CoCrO_x$ considerably exceeded other catalysts, being 10.7 times, 3.4 times, and twice as high as $FeCrO_x$, $NiCrO_x$, and $RuO_2$, respectively. Significantly, Cr doping also enhanced the current density by 192% relative to $CoO_x$ (Fig. 3b). Additionally, in order to further investigate the intrinsic activity of the catalysts, the Tafel slope was further obtained from the corresponding LSV curves (Fig. 3c). With a Tafel slope of 101.6 mV dec$^{-1}$, the $CoCrO_x$ significantly outperformed other catalysts, including the non-Cr doped $CoO_x$ catalyst (146.6 mV dec$^{-1}$) and $RuO_2$ (196.0 mV dec$^{-1}$), indicative of the more favorable OER kinetics. The electrochemical impedance spectra (EIS) (Fig. 3d) revealed that $CoCrO_x$ exhibited a lower charge transfer resistance ($R_{ct}$) in the high-frequency region (9.0 Ω) compared to $RuO_2$ (11.1 Ω), $CoO_x$ (20.1 Ω), $FeCrO_x$ (45.4 Ω), and $NiCrO_x$ (21.6 Ω), suggesting superior electrical conductivity and expedited charge transfer process. Corroborating this trend, a comparative analysis of the electrochemically active surface area (ECSA) of the catalysts (Fig. 3e and Supplementary

Fig. 14) was undertaken, obtained by measuring the double-layer capacitance in the non-faradaic potential region. The $C_{dl}$ for $CoCrO_x$ was calculated to be 16.45 mF cm$^{-2}$, higher than $FeCrO_x$ (6.13 mF cm$^{-2}$), $NiCrO_x$ (7.56 mF cm$^{-2}$), $CoO_x$ (12.12 mF cm$^{-2}$) and $RuO_2$ (9.95 mF cm$^{-2}$). Notably, both EIS and kinetic analyses are in accordance with the excellent activity of $CoCrO_x$ observed in the LSV curves. Stability is also a key criterion to evaluate the long-term catalytic performance of electrocatalysts (Supplementary Fig. 15). $CoCrO_x$ exhibited no significant loss of activity during 50 h of electrolysis in a half-cell at the constant voltage of 1.56 V (vs RHE), with the current density first increasing and then stabilizing. The increase in current density could be attributed to the formation of more active material and the gradual penetration of the electrolyte into the porous structure[23].

In order to capture the industrial potential of $CoCrO_x$ electrocatalysts, we constructed an AEM electrolyzer system with the prepared samples as an anode catalyst and commercial Pt/C as a cathode catalyst to evaluate its catalytic performance under simulated industrial conditions (60 °C). The current-voltage (I-V) characteristic curves in Fig. 3f demonstrate that the $CoCrO_x$ catalyst displays superior performances, achieving the high current density of 1.5 A cm$^{-2}$ at 2.1 V. We also investigated the impact of different membrane electrode

preparation methods on electrolyzer performance (Supplementary Fig. 16) and plotted the schematic diagrams of the electrolyzer assemblies for the three processes (Supplementary Fig. 17). To provide clarification for the outstanding performance of CoCrO$_x$ catalysts in AEMWE, the Nyquist plots of the electrolyzer obtained through EIS reveal three different resistances in AEMWEs: ohmic resistance (OR), charge-transfer resistance (CTR) and mass transport resistance (MTR) (Fig. 3g). The intercepts in the high-frequency region represents OR, while the high- and low-frequency regions arcs indicate the CTR associated with electrochemical reactions and MTR corresponding to the transportation of reactants and products, respectively. It is evident that the OR of the samples are nearly identical, with CoCrO$_x$ exhibiting the lowest CTR and MTR, indicating its superior activity derives from expedited charge transport and reduced mass transfer resistance. Supplementary Fig. 18 demonstrates that the *operando* EIS was employed at various currents to determine the charge transfer resistance of each catalyst. The obtained results indicated a notable decline in resistance with increased current, implying faster electron transfer and higher energy conversion efficiency during electrolysis at elevated currents. We additionally assessed the stability of the CoCrO$_x$ catalyst under 60 °C and room temperature conditions at a high current density of 0.5 A cm$^{-2}$ to further evaluate its potential for industrial applications (Fig. 3h and Supplementary Fig. 19). As shown in Fig. 3h, the catalyst displayed considerable stability during electrolysis durations of 120 h, verifying their practicability for real-world deployment. The diagram presented in Fig. 3h displays a diagrammatic representation of the AEMWE, which includes two stainless steel (SS) collectors with runners, two polytetrafluoroethylene (PTFE) gaskets, titanium felts, AEM, and carbon paper (CP). The various electrochemical data are compiled in Supplementary Table 2 for a cleaner comparison and contrast of each catalyst's electrochemical performance.

### Valence and local structure evolution analysis

In order to compare and investigate the key reasons for the excellent activity of CoCrO$_x$, the initial valence state of the active elements in unison is determined. By X-ray near-edge structural absorption spectroscopy (XANES), we were able to precisely determine the valence states of the active elements. For transition metal elements, the rightward shift of the nearside absorption edge represents an elevated oxidation state. Comparing the position of the absorption edge to the standard samples reveals that CoCrO$_x$ is located between CoO and Co$_3$O$_4$ (Fig. 4a). In addition, the leftward shift of the absorption edge of CoCrO$_x$ relative to CoO$_x$ indicates that the Cr doping decreases the average valence state of Co, which matches the stronger lower valence absorption peak in the soft XAS (sXAS) (Supplementary Fig. 20). The coexistence of both +2 and +3 valences for each active metal element in X-ray photoelectron spectroscopy (XPS) also confirms the above conclusion (Supplementary Fig. 21a–f and Supplementary Table 3). Correspondingly, the valence states of Cr-doped Fe and Ni (Supplementary Fig. 22) were similarly stabilized between +2 and +3 valence. Figure 4b shows that the Cr $K$-edge of all catalysts almost overlaps with Cr$_2$O$_3$, pointing to the valence state of Cr is +3, which is in high agreement with the results in sXAS (Supplementary Fig. 23) and XPS (Supplementary Fig. 21g–i). The consistency between valence and structure bolsters the reliability of further analysis. To further elucidate the reason for the difference in the intrinsic catalytic activity of the three catalysts, we examined the valence alterations in the active elements of FeCrO$_x$, CoCrO$_x$ and NiCrO$_x$ before and after the reaction using sXAS (Fig. 4c and Supplementary Fig. 24). We discovered a conspicuous change in Co alone during the reaction, its absorption peak shifted by 1.9 eV towards the higher energy, representing a significant increase in the oxidation state of Co, which is also supported by the XPS results (Supplementary Fig. 25). Given that the binding energy of Co$^{2+}$ is higher than that of Co$^{3+}$, the shift in the binding energy peak to the lower energy corroborates the higher valence

state[24,25]. It is also evident that the characteristic satellite peak of Co$^{2+}$ almost disappears, further proving that Co$^{2+}$ is almost completely oxidized to the higher valence state during the OER process.

To further uncover the dynamic evolution of the electronic structure and atomic local environment of the Co active sites during the electrocatalytic OER, the in-situ XAFS and vtc-XES measurements were performed using a lab-built electrochemical cell. As shown in Fig. 4d, the Co $K$-edge XANES spectra indicate that the positions of the absorption edge of Co under ex-situ and open-circuit voltage (OCV) conditions are almost the same. When the voltage is applied, the near-edge absorption edge of Co is positively shifted, implying the elevation of the average oxidation state of Co during the OER process, which is consistent with the sXAS and XPS results. The fitted results of the Co $K$-edge EXAFS spectra indicate that the coordination number of the first shell layer (Co-O) increases significantly from 4 to above 6 as the voltage is applied (Fig. 4e, Supplementary Fig. 26 and Supplementary Table 4), and the increase in coordination number may be the reason for the elevated Co valence state[26,27]. The graph depicted intuitively in Fig. 4f illustrates a significant positive correlation between the absorption edge position and the Co-O coordination number. In addition, we noted that the local structure of Co changed during the reaction, which may indicate the dynamic reconfiguration of CoCrO$_x$[28,29]. This is due to the partial electrolytic dissolution of Cr under alkaline conditions, which is supported by the reduction of Cr content revealed EDS mappings after reaction (Supplementary Fig. 27). The HRTEM results (Supplementary Fig. 28a) indicate that the catalyst essentially maintained its original morphology, but short lattice streaks and nanopores appeared, and these formed pores may provide a larger specific surface area and easy access to the active sites, as well as rapid mass transfer, which would further promote the OER kinetics[30,31]. The amorphous structure may have been slightly damaged during the electrolysis process, as evidenced by several bright spots in the SAED pattern after reaction. However, the original amorphous state was largely preserved (Supplementary Fig. 28b).

Although EXAFS can provide the coordination numbers as well as the metal-ligand bond lengths, it fails to distinguish between ligands with similar atomic numbers (e.g., C, N, O) or different protonation states of the ligands[32]. Therefore, the in-situ vtc-XES (Fig. 4g) was employed to identify the ligand and coordination environment of Co, thereby further confirm the ligand type and the variation in bond strength. The K$\beta$" peak around 7690 eV is significant evidence of ligand-metal interaction through σ interactions[33], and this peak is recognized as the characteristic peak of Co-O coordination[34]. An intensified K$\beta$" peak characterized implied that the Co-O interaction strengthens with voltage during the OER. This observation corroborates the contraction of the Co-O bond distance as depicted in the Fourier-transformed EXAFS (FT-EXAFS) spectra[35]. The subtracted spectra vtc-XES is illustrated on the top of Fig. 4g to highlight the differences. It is observable that post-voltage application, the peak intensity closely resembles that of the Co$_3$O$_4$ reference sample (R$_{Co-O}$ = 1.92 Å). Furthermore, Fig. 4h showcases the dependable negative connection between the intensity of the subtracted vtc-XES spectra and the Co-O bond lengths, thereby corroborating the reliability of the analysis. Shorter Co-O bonds can provide optimized binding energy for the adsorption of oxygen intermediate, thereby facilitates the accelerated OER kinetic process[36,37].

### Mechanism investigation and simulation

In order to gain deeper understanding of the adsorption behaviors and reaction mechanisms of various catalysts, we conducted in-situ SR-IR measurements and density functional theory (DFT) calculations for FeCrO$_x$, CoCrO$_x$ and NiCrO$_x$. As presented in Fig. 5a, the reflectance intensity heatmap obtained from SR-IR spectroscopy of CoCrO$_x$ indicate the three crucial reactive intermediates of OER, namely *OH (3780 cm$^{-1}$), *OOH (1014 cm$^{-1}$) and *O (930 cm$^{-1}$)[38–41]. Interestingly, the *O adsorption peaks only appear at higher potentials. When the

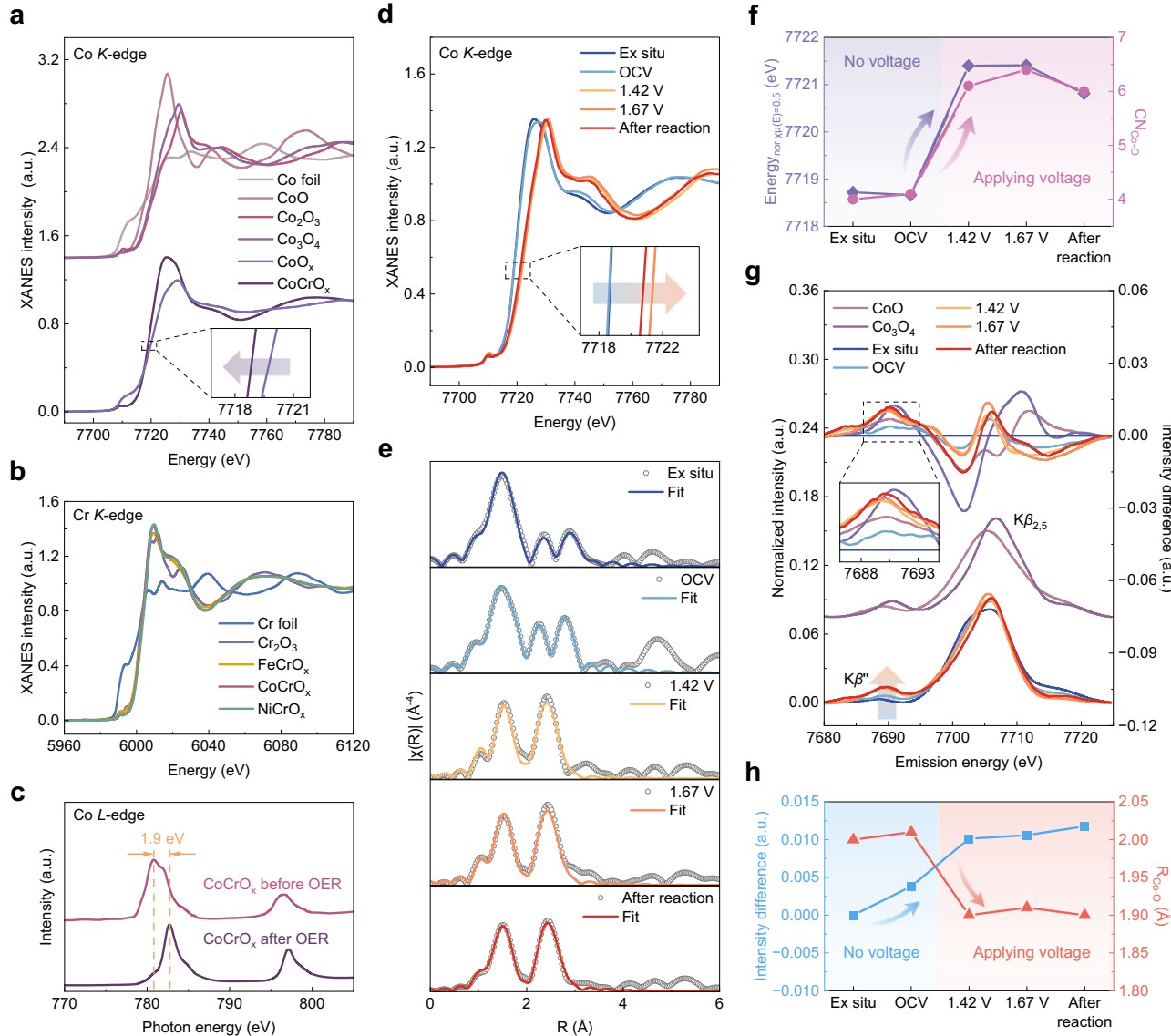

**Fig. 4 | Valence state, change in catalyst properties recorded at different applied voltages during electrocatalytic OER. a** XANES spectra of Co $K$-edge of $CoO_x$, $CoCrO_x$ and the reference samples. **b** XANES spectra of Cr $K$-edge of $FeCrO_x$, $CoCrO_x$, $NiCrO_x$ and the reference samples. **c** Normalized sXAS spectra of Co $L$-edge of $CoCrO_x$ before and after OER. **d** In-situ XANES spectra of $CoCrO_x$. **e** The fitting curves of $k^3$-weighted FT-EXAFS spectra of Co $K$-edge for $CoCrO_x$. **f** Relationship between normalized absorption edge energy [$\chi\mu(E) = 0.5$] and Co-O coordination number during the OER process. **g** In-situ vtc-XES spectra of $CoCrO_x$ and the reference samples. The subtracted vtc-XES spectra (each spectrum minus ex-situ spectrum) is shown at the top highlight the differences. **h** Relationship between intensity differences in in-situ vtc-XES spectra and Co-O bond lengths during the OER process.

potential reaches the OER region (≥1.4 V), the intensities of the *OH and *O peaks markedly increase. This indicates a positive correlation between intermediate adsorption and voltage application. Nevertheless, the intensity of the *OOH peak paradoxically weakens or even vanishes (Fig. 5b and Supplementary Fig. 29b). In contrast, no *O adsorption peaks were observed for $FeCrO_x$ and $NiCrO_x$ at all potential conditions, suggesting that $CoCrO_x$ has an optimized intermediate adsorption energy (Supplementary Fig. 29a, c). This finding further corroborates that *O is produced and adsorbed on the Co sites during the reaction of $CoCrO_x$, resulting in an increase in the number of Co-O ligands and the Co valence change, which in turn facilitates the OER process[41].

To elucidate the rationale behind these occurrence, a four-electron OER mechanism was assumed to proceed through *OH, *O and *OOH (asterisks denote adsorption sites) (Fig. 5c)[42]. The Gibbs free energy and the limiting reaction barrier were also calculated for each elementary step in the OER, which based on the free energy of the

rate-determining step (RDS) (Fig. 5d). It was discovered that for $CoCrO_x$, the RDS is the oxidation of *OH to *O, which may be a plausible explanation for the fact that *O is generated only at higher potentials. Whereas the *OOH dehydrogenation to form $O_2$ exhibits the lowest reaction energy barrier in comparison to the other three steps. As a result, the *OOH intermediates are consumed rapidly at high potentials, potentially explaining the near disappearance of the *OOH peak. It is pertinent to mention that $CoCrO_x$ exhibited the lowest RDS among three catalysts, measuring only 0.50 eV at U = 1.23 V. The RDSs for the oxidation of *O to *OOH in $FeCrO_x$ and $NiCrO_x$ were 0.76 eV and 0.63 eV, respectively. Furthermore, the differential charge densities in Fig. 5e reveals that the charge transfer of $CoCrO_x$ to *OH is significantly less and the interactions are weaker compared to $FeCrO_x$ and $NiCrO_x$. This corresponds to the relatively weaker adsorption of *OH intermediates in the $CoCrO_x$ free energy diagram. The optimized *OH adsorption allows the subsequent steps to occur at lower potentials, thus facilitating the OER kinetics[43]. Meanwhile, the total density of

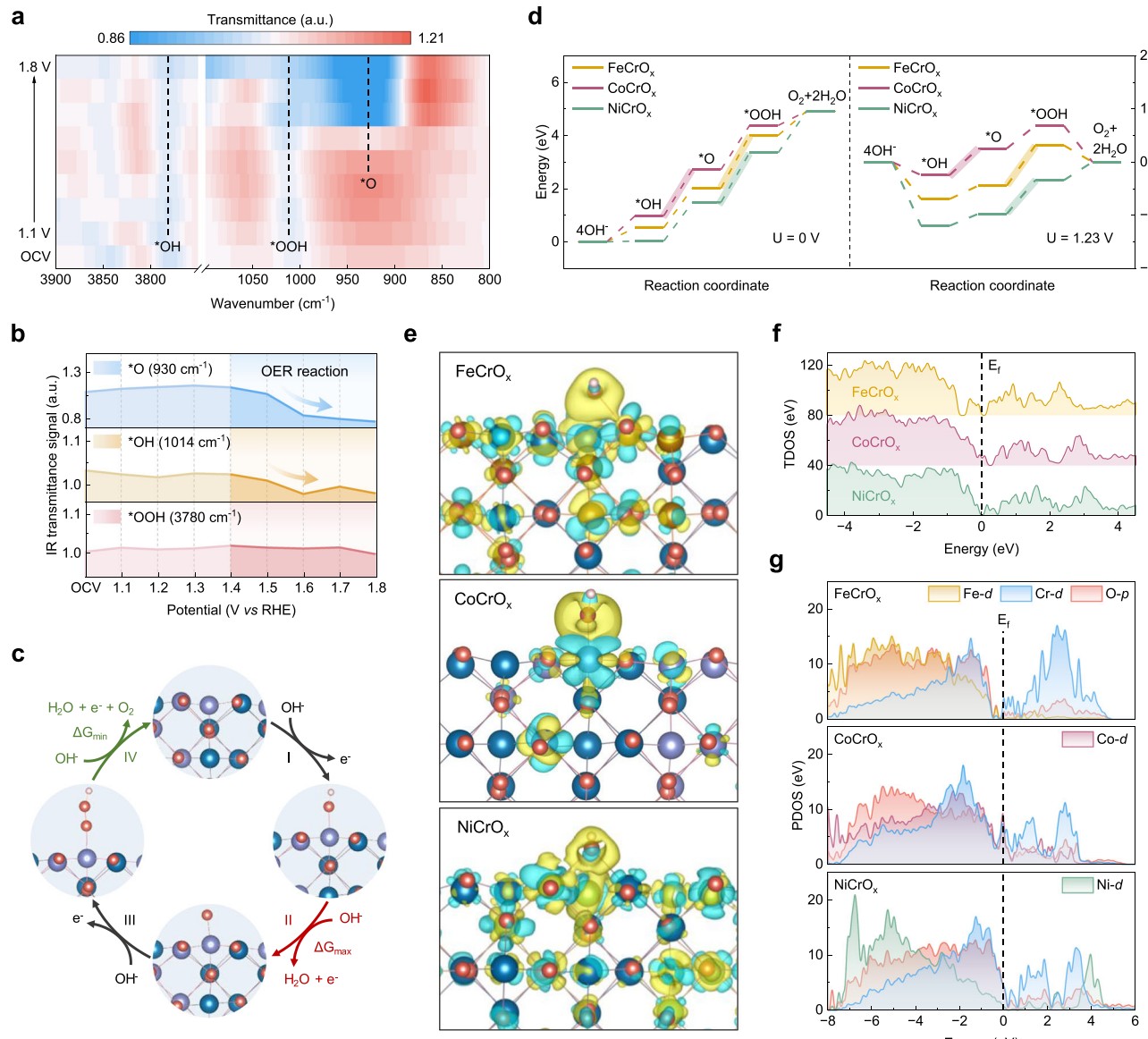

**Fig. 5 | Monitoring of reaction intermediates and DFT calculations. a** In-situ SR-IR measurements in the range of 3900–3750 cm⁻¹ and 1100–800 cm⁻¹ under various potentials for CoCrOₓ during the OER process. **b** Infrared transmittance signals versus potentials of *O, *OH and *OOH intermediates. **c** The proposed alkaline OER mechanism on CoCrOₓ. The model colors are Co (purple), Cr (blue), O (red) and H (pale pink). **d** OER free-energy diagrams at 0 V (left) and 1.23 V (right) for FeCrOₓ, CoCrOₓ and NiCrOₓ. **e** Differential charge densities (blue and yellow regions indicate electron depletion and accumulation, respectively). The model colors are Fe (yellow-brown), Co (purple), Ni (green), Cr (blue), O (red) and H (pale pink). **f** TDOS and (**g**) PDOS of FeCrOₓ, CoCrOₓ and NiCrOₓ.

states (TDOS) calculation shows that CoCrOₓ has greater density of states at the Fermi energy level, suggesting improved conductivity and explaining the exceptional activity of CoCrOₓ (Fig. 5f). The projected density of state (PDOS) illustrates that Fe/Co/Ni $d$- and O $p$-orbital electrons contributions are mainly in the valence band, whereas the conduction band is composed predominantly Cr $d$-orbital electrons (Fig. 5g). Further analysis of the $d$ orbitals of the active metal reveals that the $d_{xy}$, $d_{yz}$, $d_{z2}$, $d_{xz}$ and $d_{x2-y2}$ orbital projections of Fe, Co, and Ni mainly contribute below the Fermi level and have strong orbital interactions with the O $2p$ orbitals, indicating that Fe/Co/Ni has stable bonding ability with O atoms, which is conducive to promoting adsorption of active O atoms and OER kinetics (Supplementary Figs. 30, 31). In addition, the electron density near the Fermi energy level of the active elements is primarily contributed by the $d_{x2-y2}$ orbitals. However, the abundance of $d$ electrons near the Fermi level, rather than significantly below it, facilitates the charge transfer from

transition metals to oxygen-containing adsorbates during the OER process[44]. This is particularly evident in the case of CoCrOₓ, which is significantly higher than FeCrOₓ and NiCrOₓ. This further corroborates the reason for its exceptional activity.

## Discussion

To summarize, we elaborated and synthesized a series of high-valent Cr-doped amorphous metal oxide catalysts using a facile one-step liquid phase reduction method. The CoCrOₓ catalyst achieved a high current density of 1.5 A cm⁻² at 2.1 V and operated at a current density of 0.5 A cm⁻² for over 120 h in the AEMWE. In-situ synchrotron measurements and theoretical calculations indicate that optimized *O adsorption, shorter Co-O bond lengths, and elevated coordination numbers resulted in Co valence changes and improved OER activity. Hence, the low oxidation energy barrier of the active metal is the ultimate goal of electronic structure modulation, which is the key

factor for the greater activity of the easily oxidized transition metals. These findings broaden the pathway to enhance OER catalytic activity and practical AEMWEs performance, while the application of new in-situ method suggests numerous possibilities for addressing relevant problems in the field of energy catalytic reactions.

## Methods

### Chemicals
Iron chloride hexahydrate ($FeCl_3 \cdot 6H_2O$), nickel chloride hexahydrate ($NiCl_2 \cdot 6H_2O$), cobalt chloride hexahydrate ($CoCl_2 \cdot 6H_2O$), chromium chloride hexahydrate ($CrCl_3 \cdot 6H_2O$), sodium borohydride ($NaBH_4$) and ethanol were purchased from Sinopharm Chemical Reagent Co., Ltd. Potassium hydroxide (KOH, electronic grade, 99.999% metals basis, except sodium) was purchased from Shanghai Aladdin Biochemical Technology Co., Ltd. Ruthenium dioxide (99.9% trace metals basis) and Nafion®117 solution (5%) were purchased from Sigma-Aldrich. Carbon black (ECP600JD, Ketjenblack®) for electrochemical measurements was obtained from Suzhou Sinero Technology Co., Ltd. and graphite powder (XF009 7782-42-5) for XAS characterization was obtained from XFNANO Materials Tech Co., Ltd. All chemicals can be used without further purification.

### Synthesis of catalysts
Dissolve 11.90 g of $CoCl_2 \cdot 6H_2O$ and 13.32 g of $CrCl_3 \cdot 6H_2O$ in 25 mL of deionized (DI) water to make a 2 M metal chloride solution and set aside. Take 2 mL of the above prepared $CoCl_2$ and $CrCl_3$ solutions respectively and mix them evenly. Prepare a fresh solution of sodium borohydride by dissolving 3 g of $NaBH_4$ in 20 mL of DI water, then add dropwise to the above mixture of metal ion chlorides, stir mechanically for 15 min, allow to cool, then leave overnight and freeze dry. After freeze-drying, the catalyst was repeatedly washed using DI water and ethanol, and dried under vacuum to obtain the $CoCrO_x$ catalyst. $FeCrO_x$, $NiCrO_x$ and $CoO_x$ catalysts were obtained by the same procedure, except that the metal chloride solution was replaced with another.

### Electrochemical measurements
All electrochemical tests were performed in a typical three-electrode system on an electrochemical workstation (CHI760E, Chen Hua). A Hg/HgO electrode (1 M KOH, Tjaida) was used as the reference electrode, a platinum foil electrode ($10 \times 15 \times 0.3$) or graphite rod as the counter electrode and the catalyst was coated on $0.5 \times 0.5$ cm carbon paper (HESEN, HCP020N) to make a working electrode. 6 mg of catalyst, 3 mg of conductive carbon black (ECP600JD, Ketjenblack®) was homogenously mixed with 475 μL of DI water, 475 μL of ethanol and 50 μL of $5 wt\%$ Nafion and sonicated to make a homogeneous ink. 495 μL of DI water, 495 μL of ethanol and 10 μL of $5 wt\%$ Nafion were mixed to make $0.05 wt\%$ Nafion solution. Then 15 μL of catalyst ink was dropped onto carbon paper ($0.36$ mg cm$^{-2}$), dried and 7.5 μL of $0.05 wt\%$ Nafion was added dropwise and dried at room temperature and used for electrochemical measurements.

Linear scanning voltammetry (LSV) was performed in oxygen-saturated 1 M KOH at room temperature (25 °C) using a scan rate of 5 mV/s. The solution resistance $R_s$ was determined from the resulting Nyquist plot fit and used for ohmic drop correction based on $E_c = E_m - iR_s$ where $E_c$ is the correction potential and $E_m$ is the measured potential, respectively). EIS spectra were measured over a range of 0.1–100 kHz with an amplitude of 0.005 V. The voltage was set at 1.57 V ($vs$ RHE) and the EIS data was fitted by ZView software. In the equivalent circuit diagram (inset of Fig. 3d), $R_s$ represents the solution resistance, $R_p$ and $R_{ct}$ are the resistances for charge transfer, and $C_1$ and $C_{dl}$ are used to describe the double-layer capacitance. Electrochemical active surface area (ECSA) was obtained by measuring the double-layer capacitance in the nonfaradaic potential region with a voltage measurement range of 0.875 to 0.975 V ($vs$ RHE).

### MEAs fabrication and AEMWEs evaluation
The MEAs fabrication and AEMWEs evaluation were performed at Zhongke Enthalpy (Anhui) New Energy Technology Co, Ltd. For the anode, the prepared catalysts ($FeCrO_x$, $CoO_x$, $CoCrO_x$ and $NiCrO_x$) catalysts were mixed with the ionomer (PiperION-A5-HCO3) in an aqueous solution (1:3 water and isopropanol). For the cathode, Pt/C (TANAKA, 40 wt%) was mixed with the ionomer in an aqueous solution (1:5 water and isopropanol). The catalyst ink was sprayed onto AEM (PiperION-A60-HCO3) or platinum-titanium-coated felt (Sti0.25Pt0.5) to make gas diffusion electrodes (GDEs), with an approximate metal loading of 1 mg cm$^{-2}$ for the anode and 1.5 mgPt cm$^{-2}$ for the cathode. The AEM was immersed in 0.5 M KOH for 1 h, then replaced with fresh KOH solution and immersed for another 1 h. The bicarbonate anion was then rinsed with deionized water to convert the bicarbonate anion to hydroxide anion. The MEA was loaded into a fixture and clamped with a torque of 8 N·m. The cell was tested by a constant-potentiostat with a flowing electrolyte of 1 M KOH, and the AEMWEs were tested for internal resistance using an internal resistance meter (20 mΩ for $FeCrO_x$, 15 mΩ for $CoO_x$, 24 mΩ for $CoCrO_x$ and 22 mΩ for $NiCrO_x$). The ohmic drop was calculated based on the $E_c = E_m - iR_s$ correction for iR compensation. EIS measurements of the AEMWEs were performed by an impedance analyzer (DH7007, DongHua Analytical) over the range of 0.1–10k Hz, amplitudes of 50–250 mA, and currents of 0.5–2.5 A (The applied RMS AC is 10% of the corresponding DC and the effective area of the MEA is 5 cm$^2$). The performance and EIS tests were conducted at a temperature of 60 °C, while the durability tests were conducted at both 60 °C and room temperature.

### Material characterizations
XRD patterns were obtained by using a Philips X'Pert ProSuper diffractometer with Cu Kα radiation (λ = 1.54178 Å). The morphology was examined by SEM using the ZEISS GeminiSEM 500. TEM and HRTEM images were undertaken on a JEM-2100F field-emission electron microscope with an accelerating voltage of 200 kV. HAADF-STEM was performed on a JEOL JEM-ARF200F HRTEM with a spherical aberration corrector at voltage of 200 kV. EDS elemental mappings were obtained on JEOL JEM-F200 instrument. All catalysts for synthesis were dissolved in aqua regia and obtain the contents of Fe, Co, Ni and Cr in as-prepared catalysts were determined by ICP-AES on a PerkinElmer Optima 7300 DV ICP-AES instrument. The XPS were recorded on a Thermo ESCALAB 250Xi spectrometer with an excitation source of monochromatized Al Ka ($hv$ = 1486.6 eV) and a pass energy of 30 eV. The values of binding energies were calibrated with the C 1s peak of contaminant carbon at 284.80 eV.

### Soft X-ray XANES measurements
The $L$-edge XANES measurements of Fe, Co, Ni and Cr were performed at the photoemission endstation at BL12B beamlines of the National Synchrotron Radiation Laboratory (NSRL), China. A bending magnet was connected to the beamline, which was equipped with three gratings covering photon energies from 100 to 1000 eV with an energy resolution of ~0.2 eV. The data were recorded in the total electron yield mode by collecting the sample drain current. The resolving power of the grating was typically $E/\Delta E$ = 2000, and the photon flux was $1 \times 10^9$ photons per second.

### XAFS measurements
The XAFS spectra were measured at the BL14W1 beamline of the Shanghai Synchrotron Radiation Facility (SSRF) and the 1W1B beamline of the Beijing Synchrotron Radiation Facility (BSRF), China. The SSRF storage ring was operated at 3.5 GeV and the BSRF at 2.5 GeV, both with a maximum electron current of 250 mA. A total of about 64 mg of the appropriate sample was homogeneously mixed with graphite and pressed into round pellets of 8 mm diameter. The spectra of the $K$-edge of Cr (5989 eV), Fe (7112 eV), Co (7709 eV) and Ni (8333 eV) for

all samples were recorded in transmission mode, and the position of the absorption edge ($E_0$) was calibrated by the corresponding element foil, respectively.

The in-situ XAFS spectra were measured at the BL14W1 beamline of the SSRF, China. The in-situ XAFS measurements were performed with catalyst-coated carbon paper using a home-built electrolytic cell, the spectra of which were collected in fluorescent mode. The catalyst powders were homogeneously dispersed in water and ethanol (v:v = 1:1) to form an ink, which was dropcast onto the carbon paper as a working electrode. To obtain the evolution information of the Co sites under OER working conditions, XAFS spectra were analyzed for a series of presentative conditions (ex situ, OCV, 1.42 V $vs$ RHE and 1.67 V $vs$ RHE, after reaction).

## XAFS data analysis

The acquired EXAFS data were processed according to standard procedures using the ATHENA module implemented in the IFEFFIT software package[45]. The $k^3$-weighted $\chi(k)$ data in the $k$-space were Fourier-transformed into real (R) space using a hanning windows ($dk = 1.0$ Å$^{-1}$) to separate the EXAFS contributions from different coordination shells. To obtain detailed structural parameters around the Co atoms in the as-prepared samples, quantitative curve fitting was performed on for the Fourier-transformed $k^3\chi(k)$ in R-space using the ARTEMIS module of IFEFFIT[46]. The effective backscattering amplitude $F(k)$ and the phase shift $\Phi(k)$ of all fitting paths were calculated using the ab initio code FEFF8.0[47]. For different in-situ samples, the k range of [3.0, 10.8] Å$^{-1}$ and the $R$ range of [1.0, 3.2] Å were used for ex-situ and OCV samples. For 1.42 V $vs$ RHE, 1.67 V $vs$ RHE and after reaction samples, the $k$ range of [3.0, 11.5] Å$^{-1}$ and the $R$ range of [1.0, 3.0] Å were chosen. For the selected $R$ range for $k^3$-weighted $\chi(k)$ for the curve fitting function, the number of independent points is given by

$$N_{ipt} = \frac{2\Delta k \times \Delta R}{\pi} \tag{1}$$

As for ex-situ and OCV samples:

$$N_{ipt1} = \frac{2 \times (10.8 - 3) \times (3.2 - 1)}{\pi} = 10.9 \tag{2}$$

For 1.42 V, 1.67 V and after reaction samples:

$$N_{ipt2} = \frac{2 \times (11.5 - 3) \times (3.0 - 1)}{\pi} = 10.8 \tag{3}$$

During the curve fitting, the amplitude-efficient reduction factor ($S_0$) was fixed at the value of 0.71 for Co samples determined by fitting the data of Co foil, with the coordination number of Co-Co set to 12.

For the ex-situ and OCV samples, the FT curves showed three peaks near 1.5 Å, 2.3 Å and 2.8 Å, which were assigned to Co-O, Co-Co and Co-Cr coordination, respectively. Subsequently, a three-shell structure model including a Co-O, a Co-Co and a Co-Cr scattering path was used to fit the EXAFS data of the in-situ samples. In both samples, the Debye-Waller factors ($\sigma^2$) set to be the same, the $\sigma^2$ of the first shell layer (Co-O) of the OCV sample was set as a free parameter. Meanwhile, the $\sigma^2$ of the ex-situ sample was set to the best fitting value for the OCV sample. The coordination numbers (CN), interatomic distances ($R$), and energy shifts ($\Delta E_0$) were all set as free parameters for both samples.

Therefore, the number of adjustable parameters for the ex-situ sample is

$$N_{para1} = 3 + 3 + 3 = 9 < N_{ipt1} = 10.9 \tag{4}$$

For the OCV sample, the number of adjustable parameters is

$$N_{para2} = 4 + 3 + 3 = 10 < N_{ipt1} = 10.9 \tag{5}$$

For the 1.42 V, 1.67 V and after reaction samples, the FT curves showed two peaks near 1.5 Å and 2.4 Å assigned to Co-O and Co-Co coordination, which were fitted using a two-shell structure model including one Co-O and one Co-Co scattering path. Considering these three samples, $\sigma^2$ should be set to be consistent and CN, $R$ and $\Delta E_0$ were set as free parameters.

Hence, the number of adjustable parameters for the three samples is

$$N_{para3} = 3 + 3 = 6 < N_{ipt2} = 10.8 \tag{6}$$

All yielded R-factors for all samples were no larger than 0.020, indicating the appropriate modeling, rational parameter settings and thus, the good fitting qualities.

## Vtc-XES measurements

The Co vtc-XES measurements were measured at the BL20U beamline (E-line) of SSRF, China. The incident X-ray was monochromatized to 7706 eV by Si (111) double-crystal monochromator. Five Si (440) spherical crystals and a Pilatus3 photon counting detector were used to collect and analysis Co K$\beta_{2,5}$ and K$\beta''$ fluorescence more efficiently. To reduce the absorption of fluorescence by air, a helium gas bag was placed between the sample, spherical crystals, and the detector. The in-situ setup was identical to the in-situ XAFS measurements. The vtc-XES spectra were baseline-corrected using a straight line and normalized to the total area between 7624.6 and 7680 eV, and then the data were smoothed using the Savitzky-Golay method.

## In-situ SR-IR measurements

In-situ SR-IR measurements were conducted at the infrared beamline BL01B of NSRL using a homemade top-plate cell reflection IR setup with a ZnSe crystal as the infrared transmission window. The catalyst electrode was pressed tightly against the ZnSe crystal window with a micron-scale gap to minimize the loss of infrared light. To ensure high-quality SR-IR spectra, the apparatus utilized a reflection mode with a vertical incidence of infrared light. The infrared spectrum was obtained by averaging 128 scans at a resolution of 4 cm$^{-1}$. Prior to each systemic OER measurement, the background spectrum of the catalyst electrode was obtained at an open-circuit voltage. The OER potential ranges were measured between 1.1–1.8 V vs RHE with an interval of 0.1 V. The infrared data was processed and smoothed using OPUS software.

## DFT calculations

The calculations were performed within the Density Functional Theory (DFT) framework implanted in Vienna ab initio Simulation Package (VASP)[48]. The interaction between ions and electrons was described in the Projector Augmented Wave (PAW) Method[49]. The electron exchange and correlation energy were described using the generalized gradient approximation-based Perdew–Burke–Erzenhorf (PBE) functional[50]. The semi-empirical London dispersion corrections of Grimme and colleagues (DFT-D3) were conducted to calculate the interactions between absorbers and slabs[51]. The Hubbard-U correction was applied for better description of the localized d-electrons of Fe, Co and Ni. We chose an effective U–J value of 3.0 eV for Fe, 3.5 eV for Co and 5.5 eV for Ni atoms[52,53]. The surface structure models of FeCrO$_x$ (110), CoCrO$_x$, (110) and NiCrO$_x$ (110) were built. A sufficiently large vacuum region of 15 Å was used for all the models to ensure the periodic images were well separated. The Brillouin-zone integrations were carried out using Monkhorst-Pack grids of special points. A gamma-centered (4 × 3 × 1) k-point grid were used for supercell. To obtain the accurate structure, the plane-wave cutoff energy was set up to 500 eV.

The force convergence was set to be <0.02 eV Å$^{-1}$, and the total energy convergence was set to be <$10^{-5}$ eV. The free energy of the adsorbed state was calculated as follows based on the adsorption energy:

$$\Delta G^* = \Delta E^* + \Delta E_{ZPE} + U_{(T)} - T\Delta S \qquad (7)$$

where $\Delta E^*$ is the adsorption energy directly obtained from DFT calculations, $\Delta E_{ZPE}$ is the zero-point energy, $U_{(T)}$ is the heat capacity correction energy, and T is the temperature (T = 298.15 K), $\Delta S$ is the change in entropy. Herein, the Gibbs energy is corrected by using the VASPKIT code[54].

## Data availability
The source data underlying Figs. 2–5, Supplementary Figs. 3, 9–16, 18–26, 29–31 and the electronic structure calculations generated in this study are provided as a Source Data file. Source data are provided with this paper.

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

## Acknowledgements

This work was supported by the National Key R&D Program of China (2021YFA1600800), the National Natural Science Foundation of China (12025505, 22179125, and 12205304), the Guangdong Basic and Applied Basic Research Foundation (2022A1515011828), the Strategic Priority Research Program of the Chinese Academy of Sciences (XDB0450200), the University of China Innovation Program of Anhui Province (GXXT-2020-053), the Fundamental Research Funds for the Central Universities (WK2060000038, WK2310000113), the Youth Innovation Promotion Association CAS (2022458). We would thank NSRL, BSRF and SSRF for the synchrotron beam time and the supercomputing system in the Supercomputing Center of University of Science and Technology of China for the calculations. This work was partially carried out at the Instruments Center for Physical Science, University of Science and Technology of China.

## Author contributions

T.Y. conceived the idea and supervised the work. S.L., W.Z. and T.D. planned and performed the catalyst synthesis. S.L., M.W. and W.Z. performed the AEMWEs measurement. T.L., C.Q. and Q.L. carried out the DFT calculations. H.Z. and X.L. performed the XAFS measurements and assisted in analyzing the data. D.L., L.Z., Y.C., S.J., C.Q. and H.W. performed TEM, HRTEM, EDS-mapping measurements, and assisted in data analysis. T.Y., S.L., W.Z. and T.D. co-wrote, optimized and revised the manuscript. All authors discussed the results and provided comments on the manuscript.

## Competing interests

The authors declare no competing interests.
