## [Peer Review File · Nature Communications]

Highly efficient anion exchange membrane water electrolyzers via chromium-doped amorphous electrocatalystsREVIEWER COMMENTS

Reviewer #1 (Remarks to the Author):

This paper by Prof. Yao and co-workers reports a high-performance AEMWEs catalyst doped with high-valence metal Cr by a simple method, which is available for industrial use. They explained the critical factor for the difference in catalytic activity due to valence changes. The authors' work is systematic and comprehensive, and I believe that this finding can guide the design of catalysts for high-performance practical water electrolyzers. The manuscript is well organized and the conclusions are now well supported by the results, and I am pleased to see this manuscript published by Nature Communications after minor revisions.

1. In the materials synthesis section, the authors should explain why they chose a Co:Cr elemental ratio of 1:1 as the model catalyst for the study and whether this ratio affects catalyst performance. The authors should conduct experiments with samples of different elemental ratios to further investigate this issue.
2. Electrochemical characterization section, the authors claim to utilize operando EIS to explain the superior performance of CoCrOx. Typically, the guideline value for RMS is 5-10% of DC. If the RMS is changed, does it affect the resistance?
3. Fig. 3h, the durability test is about 2.8 V at 0.5 A cm⁻². However, the corresponding current in the I-V curve of Fig. 3f is only about 1.85 V. Please explain the reason for this discrepancy.
4. I noticed that the CoCrOx catalyst in EXAFS had a significant change in the atomic structure of Co during the OER reaction, and the authors should provide an explanation of this change in order to demonstrate this change with a possible effect on the catalytic performance.
5. I was intrigued by the authors' SR-IR explanation that the essential reason for the elevated valence of Co as the active center is the production of *O intermediates. Through careful observation, it was found that the O* absorption peak in Fig. S25b appears to be generated at around 1.5 V. However, in the in-situ XAFS, the Co valence elevation clearly occurs at 1.42 V. Please ask the authors and explain the reason for this potential difference.
6. Line 128, I noticed that the description of 'surface oxidation' might need a little revision since the synthesized material has been apparently oxidized in the bulk phase.
7. The authors should carefully check for abbreviations that are not labeled with their full name, such as XAS and XPS.

Reviewer #2 (Remarks to the Author):

The authors synthesized a series of catalysts for alkaline water electrolyzers by controllable synthesis methods, using DFT simulations combined with electrochemical characterization and a variety of in-situ spectra, they revealed the key reasons for the excellent performance of the CoCrOx catalysts in AEMWEs. After reading the manuscript as a whole, I think that the authors have carefully synthesized, tested, and analyzed the catalysts, and the characterization is nearly comprehensive and the conclusions are relatively well supported. The authors' DFT calculations seem to support the conclusions quite well. This is an excellent example of enhancing catalytic activity by modifying the valence of AEMWEs catalysts. I believe that this paper could be published in Nature Communications after the following issues could be addressed.

1. Although Fe/Co/Ni is the usual active site for OER reactions, however, Cr has likewise been shown to have some OER activity (Inorg. Chem. 2019, 58, 4014-4018), and the authors should justify the choice of the active site.
2. Fig. 5e, the authors used the differential charge analysis of the structural model of the three catalysts in the adsorption of *OH. However, they did not explain why these models were selected, which should be specifically addressed in the manuscript.
3. The authors' description of PDOS in Fig. 5g could benefit from further analysis. To better understand bonding, it would be helpful to examine the metal dx, dy, and dz orbitals. It seems like the

authors only described the contribution of Fe/Co/Ni/Cr/O to the conduction and valence bands, which might not be enough.

4. The authors should provide a more detailed description for the colors in Fig. 5e to distinguish the electron accumulation and depletion.
5. Considering the authors have conducted exhaustive experiments on the valence changes before and after OER reaction using sXAS and XPS, the authors should also provide characterization of the catalysts after electrocatalysis, such as HRTEM, in order to check whether the catalyst structure has changed significantly.
6. For XPS fitting, it is necessary to tabulate detailed parameters such as peak position and FWHM, even though the spectroscopy is presented in SI.

Other minor questions:

7. Please spell sXAS, XPS, and FT-EXAFS.
8. The range of SR-IR in the note of Fig. 5a seems to be problematic.

Reviewer #3 (Remarks to the Author):

It's commendable to see that the authors have accomplished an impressive work by developing a series of chromium-doped amorphous catalysts for alkaline water electrolysis and elucidating the key underlying mechanisms of valence changes involved therein. In particular, the catalysts synthesized by the authors exhibit remarkable activity and stability under actual electrolysis, and the explanation of their mechanism is self-consistent and nearly comprehensive. The current work is generally intriguing. I believe this manuscript is a great fit for publication in Nature Communications. There are just a few minor issues that need to be addressed.

1. In Figure 5a, the authors pointed out several key characteristic peaks in the SR-IR spectra, the authors need to reference more literature to support the peak position attribution of *OOH, *OH and *O.
2. The authors used the XPS and soft XAS tests to investigate the valence state changes of the active sites before and after the OER reaction. The reaction time should be provided to prove that the changes have been completed.
3. For in-situ XAFS and XES, please explain the reason for choosing the two voltages of 1.42 V and 1.67 V.
4. In Figure 4e, the local structure of Co is significantly changed revealed from the in-situ EXAFS spectra, it is suggested the authors to explain this change and provide relevant characterizations of the catalyst after the electrolysis reaction. These papers may be referred: Nature Communications 14.1 (2023): 4670; Angew. Chem. Int. Ed. 2023, 62 25, e202303117.
5. In Figures 3g and S17, EIS should be labeled as current density or prominently label them with membrane electrode area to avoid any potential confusion.
6. There are some minor errors in the manuscript that need to be discussed and revised:
Line 160, the panel "e" should be replaced by "d".
Lines 234 and 277, please spell "FT-XAFS".
Lines 498 and 500, "samples" should not have an "s".
7. SI, the range of SR-IR for Figure S25 is incorrect.

Overall, the authors did a good job of utilizing a variety of in-situ techniques to illustrate the connection between valence changes and catalytic performance, particularly in relation to "low oxidation energy barriers". Furthermore, the developed synthetic method here is simple and suitable for industrial-scale synthesis, and the model catalysts have exceptional activity and stability.

Response to Reviewers' Comments:

We wish to extend our sincere appreciation to all reviewers for their valuable and thoughtful comments on enhancing the quality of this manuscript. We have thoroughly examined the pertinent literature and carried out supplementary experiments to clarify the following aspects. The following is our point-by-point response to the reviewers' comments.

Our response to reviewer comments will be in blue color and the revised parts in red color, which are highlighted in yellow color in the revised Manuscript and Supplementary Information.

Reviewer #1 (Remarks to the Author):

This paper by Prof. Yao and co-workers reports a high-performance AEMWEs catalyst doped with high-valence metal Cr by a simple method, which is available for industrial use. They explained the critical factor for the difference in catalytic activity due to valence changes. The authors' work is systematic and comprehensive, and I believe that this finding can guide the design of catalysts for high-performance practical water electrolyzers. The manuscript is well organized and the conclusions are now well supported by the results, and I am pleased to see this manuscript published by Nature Communications after minor revisions.

Response: We are very grateful to Reviewer #1 for the positive comments on the manuscript and the acknowledgment of our utilization of a simple method to synthesize efficient AEMWEs catalysts. Based on your constructive comments and suggestions, we have carefully addressed these issues and revised the manuscript. In the following, we have attempted to answer all the comments made.

1. In the materials synthesis section, the authors should explain why they chose a Co:Cr elemental ratio of 1:1 as the model catalyst for the study and whether this ratio affects catalyst performance. The authors should conduct experiments with samples of different elemental ratios to further investigate this issue.

Response: We are thankful to the reviewer for this good question. As a matter of fact, during the initial screening of model catalysts, we synthesized a range of catalysts with various Co: Cr ratios ranging from 1:3 to 3:1. Our experiments revealed that the samples exhibited the optimal catalytic performance at the Co: Cr ratio of 1:1. Per your suggestion, the LSV curves of catalysts with different Co and Cr ratios have been included in the updated supplementary information, and corresponding notes have been added in the revised manuscript.

Revision:

(Page 7 of the Manuscript)

“The optimal catalytic performance of the samples was achieved when the atomic ratio of Co/Cr was 1:1, as determined through performance-based screening (Supplementary Fig. 13). For the sake of clarity, the atomic ratio of Fe/Co/Ni to Cr for the FeCrO_x, CoCrO_x and NiCrO_x catalysts is 1:1 unless otherwise specified.”

(Page S13 of the Supplementary Information)

Supplementary Fig. 13. LSV curves. LSV curves for catalysts with different Co and Cr ratios (Co: Cr =1:3~3:1).

2. Electrochemical characterization section, the authors claim to utilize operando EIS to explain the superior performance of CoCrO_x. Typically, the guideline value for RMS is 5-10% of DC. If the RMS is changed, does it affect the resistance?

Response: Thanks to the reviewers for the comments. It is true that the guideline value

for the RMS of AC is generally 5-10% of DC, but there are relatively few studies on whether AC affects Nyquist plots, and we were also curious. Therefore, we further conducted a series of experiments on CoCrO_x catalysts with different ACs applied at $\text{DC} = 1 \text{ A}$ (0.2 A cm^{-2}) and 2.5 A (0.5 A cm^{-2}). The results indicated changing the applied RMS of AC had almost no significant effect on the resistance.

Figure R1. EIS measurements. Nyquist plots of AEMWEs obtained at different direct current (DC) and alternating current (AC) for CoCrO_x as the anodes [membrane electrode assembly (MEA) area: 5 cm^2].

3. Fig. 3h, the durability test is about 2.8 V at 0.5 A cm^{-2} . However, the corresponding current in the I-V curve of Fig. 3f is only about 1.85 V . Please explain the reason for this discrepancy.

Response: We thank the reviewers for the careful observation. Indeed, both the I-V curves and the Nyquist plots were relevantly tested at $60 \text{ }^\circ\text{C}$, which is commonly used under industrial conditions. However, for the durability assessment we mainly considered the stability of the catalyst itself. Therefore, we performed the tests at room temperature ($25 \text{ }^\circ\text{C}$). Considering the possible misinterpretation, we have supplemented the durability test experiments at $60 \text{ }^\circ\text{C}$ and have changed Fig. 3h of the revised manuscript and placed the durability curves at room temperature in the SI.

Revision:

(Page 8 of the Manuscript)

(h) Chronopotentiometry curve of CoCrO_x catalyst at constant current density of 0.5 A cm^{-2} in the AEMWEs at $60 \text{ }^\circ\text{C}$. Positions indicated by the black arrows represent that the replacement of electrolyte.

(Page 10 of the Manuscript)

“We additionally assessed the stability of the CoCrO_x catalyst under $60 \text{ }^\circ\text{C}$ and room temperature conditions at a high current density of 0.5 A cm^{-2} to further evaluate its potential for industrial applications (Fig. 3h and Supplementary Fig. 19).”

(Page S19 of the Supplementary Information)

Supplementary Fig. 19. Chronopotentiometry curve. Chronopotentiometry curve of the CoCrO_x catalyst at a constant current density of 0.5 A cm^{-2} in the AEMWEs at room temperature (RT).

4. I noticed that the CoCrO_x catalyst in EXAFS had a significant change in the atomic structure of Co during the OER reaction, and the authors should provide an explanation of this change in order to demonstrate this change with a possible effect on the catalytic

performance.

Response: We thank the reviewers for their careful review. We have carefully reviewed the relevant literature and further summarized and analyzed the experimental data. It is believed that the change in Co atomic structure may be attributed to the dissolution of Cr. It is not uncommon for the dissolution of Cr to occur from transition metal catalysts when exposed to alkaline conditions (*Angew. Chem. Int. Ed.* 2023, 62, e20230985; *Electrochimica Acta* 2018, 265, 10-18; *ACS Appl. Mater. Interfaces* 2017, 9, 41239–41245). This can result in the creation of desirable pores that have the potential to increase the specific surface area and number of active sites, which may improve the catalysts performance. We also found the similar phenomenon in FeCrO_x and NiCrO_x.

5. I was intrigued by the authors' SR-IR explanation that the essential reason for the elevated valence of Co as the active center is the production of *O intermediates. Through careful observation, it was found that the O* absorption peak in Fig. S25b appears to be generated at around 1.5 V. However, in the in-situ XAFS, the Co valence elevation clearly occurs at 1.42 V. Please ask the authors and explain the reason for this potential difference.

Response: We are very thankful to the reviewers for this extremely valuable question. As is widely recognized, IR is a crucial method for monitoring reaction intermediates. It is worth noting that the potential of 1.42 V is close to the onset of the OER, and it takes a considerable amount of time for *O intermediates to be generated and accumulated. In comparison, *in-situ* XAFS requires a longer time to obtain the spectrum, while *in-situ* SR-IR is faster. Therefore, even though *in-situ* XAFS has detected the valence change at the Co site, *O still needs time to accumulate and reach the detection limit of SR-IR. This results in the characteristic peaks of *O in SR-IR appearing only at 1.5 V, and being obvious at 1.6 V. To further corroborate this analysis, we conducted three SR-IR spectra at 1.5 V with 5-minute intervals for varying reaction times. Our observations indicate a significant enhancement of the absorption peaks of *O over time, which provides compelling evidence of *O accumulation and further supports the validity of our previous analysis.

Figure R2. *In-situ* SR-IR measurements. *In-situ* SR-IR measurements for CoCrO_x at 1.5 V (vs RHE) with different reaction times.

6. Line 128, I noticed that the description of 'surface oxidation' might need a little revision since the synthesized material has been apparently oxidized in the bulk phase.

Response: Thank you for your suggestion. We have corrected 'surface oxidation' to 'oxidation' in the revised manuscript.

Revision:

(Page 7 of the Manuscript)

“The same phenomenon was observed for FeCrO_x and NiCrO_x (Supplementary Fig. 8), while the higher O content is attributed to the ease of oxidation of the transition metals.”

7. The authors should carefully check for abbreviations that are not labeled with their full name, such as XAS and XPS.

Response: We would like to express our gratitude for the reviewers' meticulous feedback and offer our sincere apologies for the oversight. We have thoroughly checked and corrected the issue in the revised manuscript.

Reviewer #2 (Remarks to the Author):

The authors synthesized a series of catalysts for alkaline water electrolyzers by controllable synthesis methods, using DFT simulations combined with electrochemical

characterization and a variety of in-situ spectra, they revealed the key reasons for the excellent performance of the CoCrO_x catalysts in AEMWEs. After reading the manuscript as a whole, I think that the authors have carefully synthesized, tested, and analyzed the catalysts, and the characterization is nearly comprehensive and the conclusions are relatively well supported. The authors' DFT calculations seem to support the conclusions quite well. This is an excellent example of enhancing catalytic activity by modifying the valence of AEMWEs catalysts. I believe that this paper could be published in Nature Communications after the following issues could be addressed.

Response: We are very grateful to Reviewer #2 for the positive comments on the manuscript and valuable suggestions on DFT calculations, your suggestions are very important to improve the quality of the manuscript. Below, we attempt to answer all the comments you have made.

1. Although Fe/Co/Ni is the usual active site for OER reactions, however, Cr has likewise been shown to have some OER activity (Inorg. Chem. 2019, 58, 4014-4018), and the authors should justify the choice of the active site.

Response: Thank you for your question. In general, Fe/Co/Ni is undoubtedly the catalytically active metal site in the OER, while Cr has relatively poor catalytic activity, so it is generally not selected as the active site (*ACS Catal.* 2015, 5, 9, 5380–5387; *Chem. Eng. J.* 2020, 402, 126144). However, during the calculations process, we still considered various adsorption active sites for rigor. It was found that the adsorption capacity of *OH intermediates on Cr atoms was too strong, which may cause difficulty in desorption of intermediates and hinder subsequent the reaction. Consequently, we chose Fe, Co and Ni as the active sites for our calculations.

Figure R2. Free-energy diagrams. Free-energy diagrams at 1.23 V for FeCrO_x, CoCrO_x and NiCrO_x.

2. Fig. 5e, the authors used the differential charge analysis of the structural model of the three catalysts in the adsorption of *OH. However, they did not explain why these models were selected, which should be specifically addressed in the manuscript.

Response: Thank you for this insightful question. The RDS for CoCrO_x with optimal catalytic activity is the oxidation of *OH to *O. The appropriate adsorption energy of *OH is crucial for the subsequent generation of key *O intermediates. Therefore, in order to maintain the unity of the research object, we chose the structural models of the three catalysts for adsorption of *OH for the study.

3. The authors' description of PDOS in Fig. 5g could benefit from further analysis. To better understand bonding, it would be helpful to examine the metal dx, dy, and dz orbitals. It seems like the authors only described the contribution of Fe/Co/Ni/Cr/O to the conduction and valence bands, which might not be enough.

Response: We appreciate the reviewers' valuable feedback. Indeed, our discussion of PDOS is somewhat limited. In response to your suggestion, we have conducted a thorough examination and analysis of the electronic orbitals of d_{xy}, d_{yz}, d_{z2}, d_{xz}, and d_{x2-y2} of the active elements Fe, Co, and Ni, investigated the essential characteristics of the catalytic activity, and further summarized the internal reaction mechanism of the catalyst.

Revision:

(Page 16-17 of the Manuscript)

“Further analysis of the d orbitals of the active metal reveals that the d_{xy}, d_{yz}, d_{z2}, d_{xz} and d_{x2-y2} orbital projections of Fe, Co, and Ni mainly contribute below the Fermi level and have strong orbital interactions with the O 2p orbitals, indicating that Fe/Co/Ni has stable bonding ability with O atoms, which is conducive to promoting adsorption of active O atoms and OER kinetics (Supplementary Figs. 30-31). In addition, the electron density near the Fermi energy level of the active elements is primarily contributed by

the $d_{x^2-y^2}$ orbitals. However, the abundance of d electrons near the Fermi level, rather than significantly below it, facilitates the charge transfer from transition metals to oxygen-containing adsorbates during the OER process⁴⁴. This is particularly evident in the case of CoCrO_x , which is significantly higher than FeCrO_x and NiCrO_x . This further corroborates the reason for its exceptional activity.”

(Page S30-31 of the Supplementary Information)

Supplementary Fig. 30. PDOS curves. PDOS of (a, b) Fe d and O 2p orbitals for FeCrO_x , (c, d) Co d and O 2p orbitals for CoCrO_x and (e, f) Ni d and O 2p orbitals for NiCrO_x .

Supplementary Fig. 31. PDOS curves. PDOS of O p orbitals for (a) FeCrO_x, (b) CoCrO_x and (c) NiCrO_x.

4. The authors should provide a more detailed description for the colors in Fig. 5e to distinguish the electron accumulation and depletion.

Response: Thank you for your suggestion, we have corrected this.

Revision: (Page 15 of the Manuscript)

“(e) Differential charge densities (blue and yellow regions indicate electron depletion and accumulation, respectively).”

5. Considering the authors have conducted exhaustive experiments on the valence changes before and after OER reaction using sXAS and XPS, the authors should also provide characterization of the catalysts after electrocatalysis, such as HRTEM, in order to check whether the catalyst structure has changed significantly.

Response: Thanks to the reviewers for this rigorous suggestion. According to your suggestion, we further supplemented the HRTEM and SAED measurements of CoCrO_x catalyst after OER. We observed that the catalyst essentially maintained its original morphology after the reaction, but a small amount of lattice streaks appeared, which can be proved by the bright spots in SAED. This may be attributed to the destruction of a small portion of the amorphous structure during the OER process.

Revision:

(Page 13 of the Manuscript)

“The HRTEM results (Supplementary Fig. 28a) indicate that the catalyst essentially

maintained its original morphology, but short lattice streaks and nanopores appeared, and these formed pores may provide a larger specific surface area and easy access to the active sites, as well as rapid mass transfer, which would further promote the OER kinetics^{30, 31}. The amorphous structure may have been slightly damaged during the electrolysis process, as evidenced by several bright spots in the SAED pattern after reaction. However, the original amorphous state was largely preserved (Supplementary Fig. 28b).”

(Page 26 of the Manuscript)

30. Xu, D. *et al.* The role of Cr doping in Ni Fe oxide/(oxy)hydroxide electrocatalysts for oxygen evolution. *Electrochim. Acta* **265**, 10-18, doi:10.1016/j.electacta.2018.01.143 (2018).

31. Bo, X., Li, Y., Hocking, R. K. & Zhao, C. NiFeCr Hydroxide Holey Nanosheet as Advanced Electrocatalyst for Water Oxidation. *ACS Appl. Mater. Interfaces* **9**, 41239-41245, doi:10.1021/acsami.7b12629 (2017).

(Page S28 of the Supplementary Information)

Supplementary Fig. 28. HRTEM image and SAED pattern. (a) HRTEM image and (b) SAED pattern of CoCrO_x after the OER process.

6. For XPS fitting, it is necessary to tabulate detailed parameters such as peak position and FWHM, even though the spectroscopy is presented in SI.

Response: Thank you for this suggestion. The presentation of the XPS fitting details is necessary for rigorous reasons. We have summarized these results into a table and

incorporated them into the SI.

(Page S34 of the Supplementary Information)

Supplementary Table 3. The detailed XPS fitting results of Supplementary Fig. 21.

Catalyst	Element	Peak	Position (eV)	FWHM (eV)	Area
FeCrO _x	Fe	Fe ²⁺ 2p _{3/2}	710.9	3.0	10522.6
		Fe ³⁺ 2p _{3/2}	713.6	3.3	5442.0
		Fe 2p _{3/2} sat.	719.3	5.8	5747.3
		Fe ²⁺ 2p _{1/2}	724.5	3.0	5261.3
		Fe ³⁺ 2p _{1/2}	727.3	3.3	2721.0
		Fe 2p _{1/2} sat.	732.5	5.8	2771.2
	Cr	Cr ³⁺ 2p _{3/2}	577.3	2.5	29268.1
		Cr ³⁺ 2p _{1/2}	587.0	2.5	11556.0
CoCrO _x	Co	Co ³⁺ 2p _{3/2}	780.5	2.1	6538.0
		Co ²⁺ 2p _{3/2}	781.8	3.0	12074.5
		Co 2p _{3/2} sat.	785.9	6.5	15082.8
		Co ³⁺ 2p _{1/2}	796.3	2.1	3533.2
		Co ²⁺ 2p _{1/2}	797.6	3.0	7719.5
		Co 2p _{1/2} sat.	803.0	5.1	6769.3
	Cr	Cr ³⁺ 2p _{3/2}	577.2	2.6	24307.8
		Cr ³⁺ 2p _{1/2}	587.0	2.6	9625.5
NiCrO _x	Ni	Ni ²⁺ 2p _{3/2}	855.9	2.2	12442.1
		Ni ³⁺ 2p _{3/2}	857.1	2.5	6329.0
		Ni 2p _{3/2} sat.	861.7	5.0	16945.9
		Ni ²⁺ 2p _{1/2}	873.5	2.2	6317.2
		Ni ³⁺ 2p _{1/2}	875.5	2.5	3279.3
		Ni 2p _{1/2} sat.	879.7	5.0	11624.1
	Cr	Cr ³⁺ 2p _{3/2}	577.2	2.3	22617.6
		Cr ³⁺ 2p _{1/2}	586.8	2.3	10399.5

7. Please spell sXAS, XPS, and FT-EXAFS.

Response: We thank the reviewers for the careful advice, and we apologize for this oversight; we have added the full name of the above abbreviation to the revised manuscript.

8. The range of SR-IR in the note of Fig. 5a seems to be problematic.

Response: We would like to thank the reviewers for providing careful advice, and we have corrected this error.

Revision:

(Page 15 of the Manuscript)

“(a) *In-situ* SR-IR measurements in the range of 3900-3750 cm^{-1} and 1100-800 cm^{-1} under various potentials for CoCrO_x during the OER process.”

Reviewer #3 (Remarks to the Author):

It's commendable to see that the authors have accomplished an impressive work by developing a series of chromium-doped amorphous catalysts for alkaline water electrolysis and elucidating the key underlying mechanisms of valence changes involved therein. In particular, the catalysts synthesized by the authors exhibit remarkable activity and stability under actual electrolysis, and the explanation of their mechanism is self-consistent and nearly comprehensive. The current work is generally intriguing. I believe this manuscript is a great fit for publication in Nature Communications. There are just a few minor issues that need to be addressed.

Response: We are very grateful to Reviewer #3 for the positive comments and valuable suggestions, which have led to substantial improvement of the manuscript. Next, we have attempted to answer all the comments you have made.

1. In Figure 5a, the authors pointed out several key characteristic peaks in the SR-IR spectra, the authors need to reference more literature to support the peak position attribution of *OOH, *OH and *O.

Response: Thank you very much for your valuable suggestion and we have cited additional literatures to further support the reliability of peak attribution. It is well known that *OOH, *OH and *O are the three key reaction intermediates of OER. Accurately identifying them is crucial in revealing the valence change and mechanism of CoCrO_x catalysts, which exhibit excellent activity.

Revision:

(Page 14 of the Manuscript)

“As presented in Fig. 5a, the reflectance intensity heatmap obtained from SR-IR spectroscopy of CoCrO_x indicate the three crucial reactive intermediates of OER, namely *OH (3780 cm⁻¹), *OOH (1014 cm⁻¹) and *O (930 cm⁻¹)³⁸⁻⁴¹.”

(Page 26-27 of the Manuscript)

38. Su, H. *et al.* Dynamic Evolution of Solid–Liquid Electrochemical Interfaces over Single-Atom Active Sites. *J. Am. Chem. Soc.* **142**, 12306-12313, doi:10.1021/jacs.0c04231 (2020).

39. Su, H. *et al.* In-situ spectroscopic observation of dynamic-coupling oxygen on atomically dispersed iridium electrocatalyst for acidic water oxidation. *Nat. Commun.* **12**, 6118, doi:10.1038/s41467-021-26416-3 (2021).

40. Lin, C. *et al.* In-situ reconstructed Ru atom array on α -MnO₂ with enhanced performance for acidic water oxidation. *Nat. Catal.* **4**, 1012-1023, doi:10.1038/s41929-021-00703-0 (2021).

41. Zhou, Z. *et al.* Cation-Vacancy-Enriched Nickel Phosphide for Efficient Electrosynthesis of Hydrogen Peroxides. *Adv. Mater.* **34**, 2106541, doi:10.1002/adma.202106541 (2022).

2. The authors used the XPS and soft XAS tests to investigate the valence state changes of the active sites before and after the OER reaction. The reaction time should be provided to prove that the changes have been completed.

Response: Thank you for your constructive suggestions. In our testing of the relevant characteristics of the after-reaction samples, we recorded the corresponding chronoamperometry (i-t) curves for a reaction time of approximately 6000 seconds. We

observed that the current density of the FeCrO_x, CoCrO_x, and NiCrO_x samples displayed a similar trend. The current density of these samples initially increased and then leveled off. This could be attributed to the formation of more active material from the catalyst and the gradual penetration of the electrolyte into the porous structure. Therefore, the phenomenon of current density leveling off can be interpreted as evidence that the catalyst change has been completed and the collected characterization data are credible.

Figure R3. I-t curves. Chronoamperometry curves of FeCrO_x at 1.67 V, (b) CoCrO_x at 1.54 V and (c) NiCrO_x at 1.61 V (vs RHE).

3. For in-situ XAFS and XES, please explain the reason for choosing the two voltages of 1.42 V and 1.67 V.

Response: Thanks for the constructive question. The initial motivation of our selection of the two typical voltages was to identify representative regions in the OER process. The voltage of 1.42 V is situated close to the onset voltage the OER, while 1.67 V falls within the region of intense reaction. The two voltages were selected to monitor the changes in electronic and atomic structure of the reactive sites, providing a representative overview.

4. In Figure 4e, the local structure of Co is significantly changed revealed from the in-situ EXAFS spectra, it is suggested the authors to explain this change and provide relevant characterizations of the catalyst after the electrolysis reaction. These papers may be referred: Nature Communications 14.1 (2023): 4670; Angew. Chem. Int. Ed.

2023, 62 25, e202303117.

Response: Thank you for your valuable feedback and offering high-quality literature. We have supplemented the corresponding discussions in the revised manuscript and supporting information, and have cited the aforementioned references at appropriate locations.

Revision:

(Page 13 of the Manuscript)

“In addition, we noted that the local structure of Co changed during the reaction, which may indicate the dynamic reconfiguration of CoCrO_x ^{28, 29}. This is due to the partial electrolytic dissolution of Cr under alkaline conditions, which is supported by the reduction of Cr content revealed EDS mappings after reaction (Supplementary Fig. 27). The HRTEM results (Supplementary Fig. 28a) indicate that the catalyst essentially maintained its original morphology, but short lattice streaks and nanopores appeared, and these formed pores may provide a larger specific surface area and easy access to the active sites, as well as rapid mass transfer, which would further promote the OER kinetics^{30, 31}. The amorphous structure may have been slightly damaged during the electrolysis process, as evidenced by several bright spots in the SAED pattern after reaction. However, the original amorphous state was largely preserved (Supplementary Fig. 28b).”

(Page 26 of the Manuscript)

28. Lv, L. *et al.* Coordinating the Edge Defects of Bismuth with Sulfur for Enhanced CO_2 Electroreduction to Formate. *Angew. Chem. Int. Ed.* **62**, e202303117, doi:10.1002/anie.202303117 (2023).

29. Zhu, J. *et al.* Surface passivation for highly active, selective, stable, and scalable CO_2 electroreduction. *Nat. Commun.* **14**, 4670, doi:10.1038/s41467-023-40342-6 (2023).

30. Xu, D. *et al.* The role of Cr doping in Ni Fe oxide/(oxy)hydroxide electrocatalysts for oxygen evolution. *Electrochim. Acta* **265**, 10-18, doi:10.1016/j.electacta.2018.01.143 (2018).

31. Bo, X., Li, Y., Hocking, R. K. & Zhao, C. NiFeCr Hydroxide Holey Nanosheet as

Advanced Electrocatalyst for Water Oxidation. *ACS Appl. Mater. Interfaces* **9**, 41239-41245, doi:10.1021/acsami.7b12629 (2017).

(Page S27-28 of the Supplementary Information)

Supplementary Fig. 27. STEM image and corresponding EDS mappings. STEM image and corresponding EDS mappings of CoCrO_x after the OER process showing the dispersion of Co (pink), Cr (blue-green) and O (yellow), respectively.

Supplementary Fig. 28. HRTEM image and SAED pattern. (a) HRTEM image and (b) SAED pattern of CoCrO_x after the OER process.

5. In Figures 3g and S17, EIS should be labeled as current density or prominently label them with membrane electrode area to avoid any potential confusion.

Response: Thank you for your detailed suggestions. We have modified Fig. 3g and Supplementary Fig. 17 and labeled the membrane electrode area in the figure notes to avoid any possible misunderstanding.

Revision:

(Page 8 of the Manuscript)

Fig. 3. Electrocatalytic activity and stability. (g) Nyquist plots of AEMWEs applying 1 A direct current (DC) and 100 mA root mean square (RMS) alternating current (AC) [membrane electrode assembly (MEA) area: 5 cm²].

(Page S18 of the Supplementary Information)

Supplementary Fig. 18. Operando EIS measurements. Nyquist plots of AEMWEs obtained at different currents for (a) FeCrO_x, (b) CoO_x, (c) CoCrO_x and (d) NiCrO_x as the anodes (MEA area: 5cm²).

6. There are some minor errors in the manuscript that need to be discussed and revised:

Line 160, the panel "e" should be replaced by "d".

Lines 234 and 277, please spell "FT-XAFS".

Lines 498 and 500, "samples" should not have an "s".

Response: We appreciate your scrutiny and we apologize for the error. We have corrected the above mistakes.

7. SI, the range of SR-IR for Figure S25 is incorrect.

Response: Thank you very much for pointing out our error, we apologize for it and have corrected it in the revised supplementary information.

Revision:

(Page S29 of the Supplementary Information)

Supplementary Fig. 29. *In-situ* SR-IR measurements. *In-situ* SR-IR measurements in the range of 3900-3700 cm^{-1} and 1150-800 cm^{-1} under various potentials for (a) FeCrO_x , (b) CoCrO_x and (c) NiCrO_x during the OER process.

Overall, the authors did a good job of utilizing a variety of in-situ techniques to illustrate the connection between valence changes and catalytic performance, particularly in relation to "low oxidation energy barriers". Furthermore, the developed synthetic method here is simple and suitable for industrial-scale synthesis, and the model catalysts have exceptional activity and stability.

Response: The "low oxidation energy barrier" is the critical factor for the valence change leading to high catalytic activity, and a simple and readily available synthetic method is inevitably required for industrial applications. Thank you again for your kind recognition of our research work.

REVIEWERS' COMMENTS

Reviewer #1 (Remarks to the Author):

After the review, this work could be accepted.

Reviewer #2 (Remarks to the Author):

I would like to recommend this paper because the authors have answered all my questions

Reviewer #3 (Remarks to the Author):

The authors have addressed the comments/suggestions carefully. I am satisfied with the revisions. This manuscript is now acceptable for publication.